# Suppression of Alkalization in Rainwater Regulating Reservoir by Shading on a Pilot Scale

**Hiroshi Asakura [1],\*, Umio Matsuse [2] and Kei Nakagawa [1]**

1 Institute of Integrated Sciences and Technology, Nagasaki University, Nagasaki 852-8521, Japan; kei-naka@nagasaki-u.ac.jp
2 Sankyo landfill Site, Department of Environment, Nagasaki 850-8685, Japan; matsuse_umio@yahoo.co.jp
\* Correspondence: asakura_hiroshi@yahoo.co.jp

**Abstract:** As water in a rainwater regulating reservoir at the Sankyo landfill site in Nagasaki City tends to be alkalized and to exceed the pH upper limit of 7.5, measures to suppress the alkalization should be implemented. Inhibiting photosynthesis in algae is required to suppress the alkalization. Shading is one of the methods for inhibiting algal photosynthesis. In this study, we evaluated the pH reduction effect of shading on a pilot scale. pH decreased from 7.28 to 7.15 when 3% of the total area of the rainwater regulating reservoir was shaded. In addition, a clear decrease in pH was observed with more than 60% shading.

**Keywords:** landfill site; regulating reservoir; alkalization; shading; algae

## 1. Introduction

High pH has been reported in lakes, rivers and waterways, and it is assumed that photosynthesis in algae is one of the causes of this phenomenon [1–7]. Photosynthesis promotes the absorption of inorganic carbon in water (IC, dissolved $CO_2$) by algae, which alkalizes the water [1,8,9]. $CO_2$ and carbonate ions in water are in equilibrium with $CO_2$ in air, as shown in Equations (1) and (2). During photosynthesis in algae, $CO_2$ in water is consumed, causing the reaction in Equation (2) to shift to the left, which decreases $H^+$ and results in an increase in pH [10,11].

$$CO_2 \text{ (g)} \rightleftarrows CO_2 \text{ ($\ell$)} \tag{1}$$

$$CO_2 \text{ ($\ell$)} + H_2O \rightleftarrows H_2CO_3 \rightleftarrows H^+ + HCO_3^- \tag{2}$$

At the Sankyo landfill site (N 32.83735, E 129.77051), which is a final disposal site for municipal solid waste in Nagasaki City, the tendency of alkalization is high in the rainwater regulating reservoir (RRR) in summer, and the water turns green. We believe that one of the causes of alkalization is photosynthesis in blue-green algae. In Japan, national effluent standards apply to discharge into public waters such as rivers. The water in the Sankyo RRR is neutralized with chemicals, resulting in long-term treatment costs. Therefore, it is necessary to develop a low-cost method to control alkalization in the long term.In order to suppress alkalization, it is necessary to inhibit algal growth and photosynthesis. Measures to inhibit algal growth in closed water bodies include aeration, removal, ultraviolet light treatment, electric and magnetic field treatment and shading [12]. Shading causes algae to self-consume and decrease [13]. As the objective of the Sankyo RRR is to reduce the cost of chemicals for neutralization, we avoided methods that incur an ongoing cost and selected shading.

In our previous work [14], we evaluated the relationship between the shading effect and the alkalization suppression effect by applying several shading materials to a large outdoor water tank filled with water from the Sankyo RRR, and found that the light quantum should be kept below approximately 20 μmol/(m²s). In addition, we proposed a

low-cost shading material such as a shading cover. Conventional methods of inhibiting algal blooms in water storage tanks are based on the use of closed shading to prevent light from reaching the water surface [12,15]. It has been reported that abnormal algal growth can be inhibited by covering 30–60% of the water surface with a plastic floating plate [16]. Lowering water temperature by shading also inhibits algal growth [12]. In a reservoir in Los Angeles, black plastic balls (shade balls) are floated to shade the water [17]. In the laboratory, shading reduced algal blooms [18,19]. In pilot-scale experiments (1 to 10 m$^2$ for approximately one week), chlorophyll decreased and pH decreased as well [20,21].

From the above, we considered that if the shading effect were high, algal growth would be reduced, the decrease in IC by photosynthesis would be inhibited, and subsequently, alkalization would be suppressed. In order to control the alkalization of the RRR, Sankyo is planning to shade the RRR based on our previous report [14]. Determining the shading ratio and the alkalization suppression effect in the actual RRR is crucial for field application.

In this study, we conducted a pilot-scale shading experiment and evaluated the pH reduction effect in a RRR.

## 2. Water Alkalization Problem in Rainwater Regulating Reservoir at Sankyo Landfill Site

At the Sankyo landfill site, incinerated ash of combustible waste and the residue obtained after recycling incombustible and bulky waste from households are landfilled. The RRR at the landfill site stores rainwater at and around the site and controls the discharge of rainwater to the downstream area. Therefore, rainwater that has washed over the soil in the site is collected in the RRR by gutters, and not seepage water from the landfilled waste (called leachate). In other words, there is no relationship between the waste and the water in the RRR. Leachate, on the other hand, is collected in a leachate regulating reservoir and treated by biological oxidation, coagulative precipitation, sand filtration and activated carbon adsorption.

The RRR has a catchment area of 1,080,000 m$^2$, a regulating reservoir area of 22,430 m$^2$ and a volume of 204,000 m$^3$. By simply dividing the volume by the area, we obtain the estimated depth of approximately 9 m. The effluent from this reservoir is mixed with treated leachate and discharged into the nearby Mie River. At Sankyo, water quality is regularly measured at the surface layer and the middle layer of the RRR near the reservoir outlet and at the final effluent immediately before the confluence of the Mie River. The sampling points for regular measurements are shown in Figure 1 as SP. Surface water and final effluent are sampled within 50 cm from the water surface with bucket, and middle layer water is sampled with Heyroth water sampler at a depth of several meters from the water surface near the reservoir outlet. The depth of the middle layer cannot be clarified because the position of the outlet is fixed while the water level changes with the season.

In Japan, national effluent standards apply to discharge into public waters such as rivers. The permissible limit for pH is 5.8–8.6, and this is applied to the final effluent in Figure 1. Local residents often oppose the construction of landfill sites. In order to build a consensus between the local residents and the builder, sometimes, standards that are stricter than the national effluent standards are set. For this reason, the effluent standards for pH of 6.0–7.5 were set (agreed value with local residents). However, it should be noted that there is no scientific basis for these standards, i.e., these are only nice round numbers. As the pH of RRR water is high in summer, there is concern that the pH of the final effluent may exceed the agreed value. To this end, measures are being taken to lower the pH by treating the leachate with sulfuric acid, thereby lowering the pH of the final effluent after mixing with the effluent from the RRR. On average, the pH of the treated leachate was lowered to 6.8 from 2014 to 2018 and the pH of the final effluent was below the upper limit of 7.5. However, the use of sulfuric acid has increased the cost of leachate treatment, and an enormous cost will be incurred if the treatment is continued for more than 80 years until the site is decommissioned. Therefore, an inexpensive and effective method to reduce the pH of the RRR is needed.

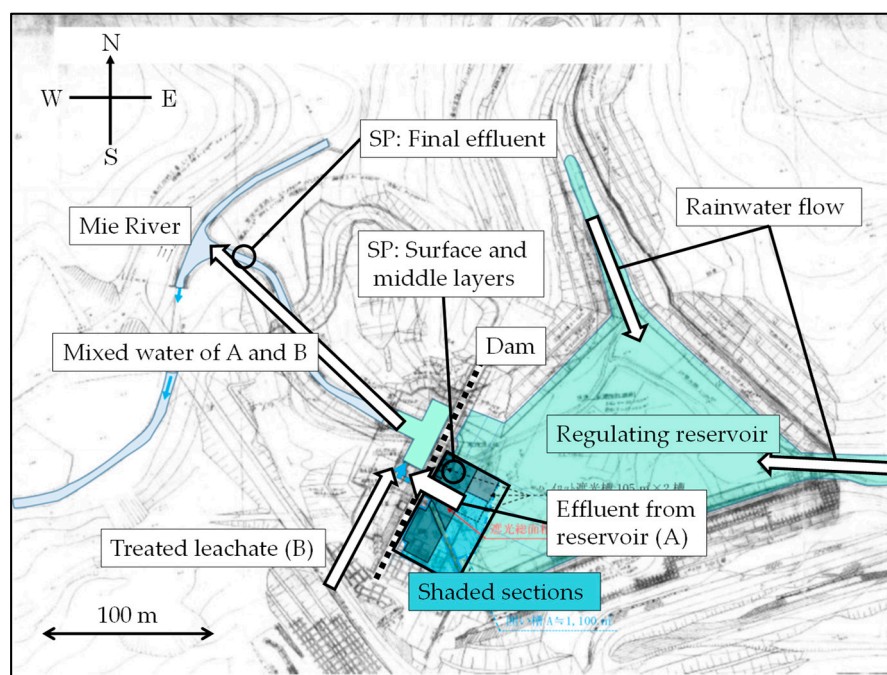

**Figure 1.** Direction of water flow and water sampling points for regular measurements in Sankyo RRR.

The seasonal trend of pH (TOADKK, MM-41DP) of the surface and middle layers of the RRR from 2012 to 2020 is shown in Figure 2a. For the surface layer, the pH is high during the summer months of June to September, particularly in June, when the average pH exceeds 8. The pH of the middle layer is lower than that of the surface layer, and the average pH in June is slightly above the upper limit of 7.5.

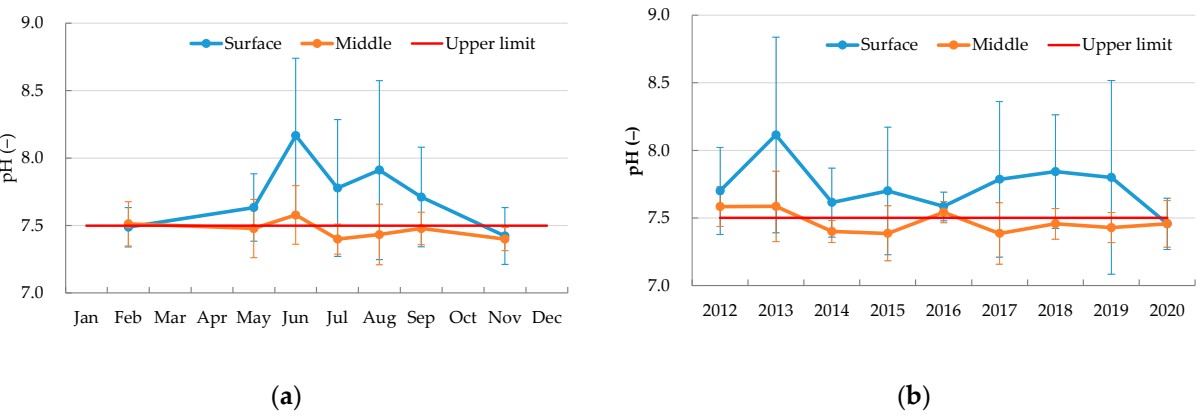

(a)                                                                                                            (b)

**Figure 2.** Average pH of Sankyo RRR: (**a**) seasonal trend (2012–2020); (**b**) secular trend; error bar: SD.

The secular trend of pH in the surface and middle layers from 2012 to 2020 is shown in Figure 2b. The pH of the surface layer is higher than that of the middle layer. The pH of the middle layer is almost near the upper limit of the agreed value. No significant secular trend is observed.

As the effluent from the RRR is from the middle layer, it is necessary to lower the pH of the middle layer in order to lower the pH of the effluent. Photosynthesis is likely to be more active in the surface layer where light reaches, and it is easier to collect water from the surface layer for the experiment. We assume that there is a relationship between the surface layer pH and the middle layer pH. In that case, the surface layer pH could be examined instead of the middle layer pH, and this would simplify the experiment. Figure 3 shows

the correlation between the surface layer pH and the middle layer pH of the RRR from 2012 to 2020. As there is moderate correlation between the two, it is possible that the trend of the middle layer pH can be determined by examining the surface layer pH.

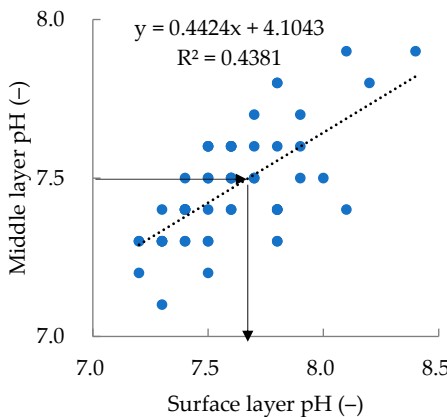

**Figure 3.** Correlation between surface and middle layer pHs of Sankyo RRR (2012–2020).

## 3. Materials and Methods

### 3.1. Outline

In the Sankyo RRR, a part of the surface layer near the reservoir outlet was divided into several sections, and shading material was applied at various shading ratios. The trend of water quality change was monitored to evaluate the effect of shading on pH reduction. The study period was from 19 June 2020 to 5 February 2021.

### 3.2. Methods

Shading material was prepared by processing a weed prevention sheet (Tanaka Co., Ltd., Otsu, Osaka, Japan; JY-200) that inhibits weed growth by shading. The material is black nonwoven polyester fabric having a unit size of 2 m (width) $\times$ 100 m (length) $\times$ 0.5 mm (thickness), a unit weight of 200 g/m$^2$ and a shading rate of 99.9%. The edges of the sheet were folded and sewn into a cylinder, and waste 2 L plastic bottles were placed inside the cylinder to provide buoyancy by acting as a float. As an alternative, many bottles were tied to the underside of the sheet. On a clear day, when light quantum was 1807 $\mu$mol/(m$^2$s), that under the sheet within 5 cm was 0 $\mu$mol/(m$^2$s). The photosynthesis activity increased linearly up to approximately 200 $\mu$mol/(m$^2$s) [22,23]. It was reported that the specific growth rate of algae increased up to a quantum density of approximately 5 mol/(m$^2$d) (=580 $\mu$mol/(m$^2$s)) [24]. In our previous report [14], we proposed that keeping the light quantum below approximately 20 $\mu$mol/(m$^2$s) would effectively control alkalization. Therefore, the shading material used in this study has sufficient shading effect.

The RRR and the shaded sections are shown in Figure 1. The pH reduction effect with and without the shading material and with different shading areas was evaluated. The shaded sections were classified into three categories: large section, small section and entire shaded section. The shading effect was more practically understood in the large section. However, because the disturbance factor was too large for a large-scale experiment and might result in experimental failure, we also used the small sections to determine the shading effect precisely. We assumed that the pH differed between the inside and outside of the entire shaded section.

First, there are two large sections: the unshaded section (LN: large not covered) and the shaded section (LC: large covered), each measuring approximately 10 m on one side. The pH of LC would be lower than that of LN. The small sections consist of squares of approximately 2.4 m on one side, with stepwise varying percentages of the shaded area: 0% (S00), 40% (S40), 60% (S60), 80% (S80) and 100% (S100). The pH decreases as the percentage of shaded area increases, and is lowest most likely at 100% (S100). Finally, to

compare the cases with and without shading, there is outside the shaded section (OUT) and runoff after passing through the entire shaded section (effluent from entire shaded section, EFL). EFL will have a lower pH than OUT. The large and small sections have a 2 m deep curtain underneath the enclosure, and the lower part is open, i.e., water flux would be low. A schematic diagram of the shaded sections and a photograph are shown in Figure 4a,b, respectively. The shading conditions are shown in Table 1. The cost of installing the shading material is 0.7 million yen for materials and 4.3 million yen for construction, totaling approximately 5 million yen (0.0077 euro/yen, August 2021). The shaded area measures approximately 700 m$^2$, and the percentage of shaded area in the RRR is approximately 3% (=700 m$^2$/22,430 m$^2$). The percentage of shaded area was determined by the budget ceiling and the restrictions brought about by using an actual facility for the experiment, and values in the literature were not quoted.

Water samples were collected approximately once a week. First, we measured the open-air light quantum at the site. A fishing rod, a reel, a fishing line and a stainless-steel water bottle were used as water sampling equipment (Figure 4c). Guide lines were laid out from the fence on the dam to each shaded section, and the water bottle to which the fishing line was connected was dipped into the guide line. The reel was loosened to send the water bottle to the water surface to fill it with water. Then the reel was reeled, and the water bottle was pulled up to collect water. Sheets of LC and S100 were temporarily flipped up.

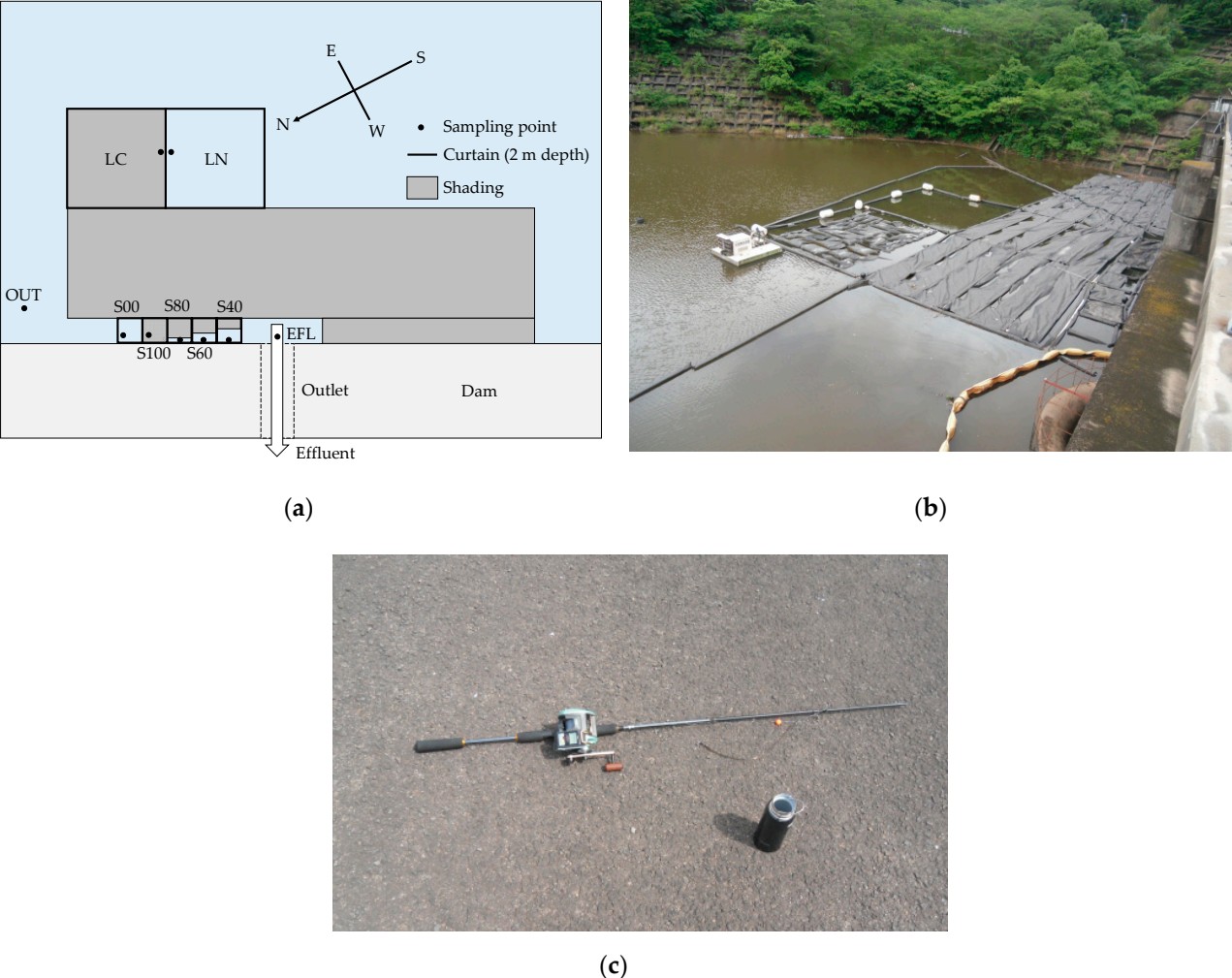

**Figure 4.** (**a**) Schematic diagram of shaded sections; (**b**) Photograph of the sections (direction: from OUT to LN in (**a**)); (**c**) Water sampling equipment.

**Table 1.** Shading conditions.

| Category | Abbreviation | Description |
|---|---|---|
| Large section | LN | non-shaded |
| | LC | shaded |
| Small section | S00 | 0% shaded |
| | S40 | 40% shaded |
| | S60 | 60% shaded |
| | S80 | 80% shaded |
| | S100 | 100% shaded |
| Entire section | OUT | Outside shaded section |
| | EFL | Effluent from entire shaded section |

Water temperature, pH, electrical conductivity (EC), oxidation-reduction potential (ORP), dissolved oxygen (DO) and chlorophyll were measured immediately at the site. After the measurements, 100 mL polyethylene bottles were filled with water and stored at ambient temperature for maximum 2 h. Open-air light quantum was measured after all the water samples were collected. The sampled water in the bottles was brought back to the laboratory and filtered through a 0.45 μm pore size filter. Then, IC, total nitrogen and total phosphorus were measured. The obtained pH was statistically processed using the Wilcoxon signed-rank test and the Friedman test.

*3.3. Measurement Equipment*

Light quantum (Apogee, SE-MQ-200), water temperature (Tanita, TT-508N), pH (HORIBA, B-212), EC (HORIBA, B-173), ORP (CUSTOM, PH-6600), DO (SATO, DO-5509) and chlorophyll (Kasahara Chemical Industry, Kuki, Saitama, Japan; Chlorophyll Sensor CHL-30, uranine-equivalent fluorescence intensity) were measured. IC (Shimadzu, TOC-VWS) and total nitrogen and total phosphorus (HACH, DR2700) were measured in the filtrate.

## 4. Results

*4.1. Average Values of Water Quality Parameters*

The average values of water quality parameters for the whole experiment are shown in Table 2.

**Table 2.** Average values of water quality parameters ($n = 27$).

| | | Large section | | Small section | | | | | Entire section | |
|---|---|---|---|---|---|---|---|---|---|---|
| | | LN | LC | S00 | S40 | S60 | S80 | S100 | OUT | EFL |
| Temp. | °C | 20.1 | 20.7 | 20.3 | 20.2 | 20.1 | 20.1 | 20.1 | 20.4 | 20.4 |
| pH | – | 7.16 | 7.13 | 7.19 | 7.13 | 7.09 | 6.96 | 6.87 | 7.28 | 7.15 |
| EC | μS/cm | 585 | 584 | 577 | 564 | 575 | 571 | 567 | 583 | 574 |
| ORP | mV | 210 | 227 | 262 | 260 | 257 | 254 | 245 | 266 | 267 |
| DO | mg-$O_2$/L | 9.0 | 8.5 | 8.5 | 8.5 | 8.4 | 8.1 | 7.5 | 8.6 | 8.3 |
| Chlorophyll | μg/L | 4.9 | 8.6 | 4.9 | 7.4 | 7.6 | 15.5 | 13.0 | 4.4 | 5.8 |
| TOC | mg-C/L | 5.5 | 4.9 | 5.1 | 5.0 | 4.8 | 4.7 | 4.7 | 5.2 | 5.1 |
| IC | mg-C/L | 6.0 | 5.8 | 5.5 | 5.0 | 5.2 | 5.0 | 4.8 | 5.9 | 6.0 |
| T-N | mg-N/L | 2.9 | 3.5 | 3.7 | 4.1 | 3.9 | 3.9 | 3.8 | 4.3 | 3.7 |
| T-P | mg-P/L | 0.17 | 0.14 | 0.16 | 0.13 | 0.13 | 0.12 | 0.17 | 0.13 | 0.12 |

*4.2. Weather*

Temperature and precipitation (Japan Meteorological Agency) are shown in Figure 5a. Daily average temperature exceeded 25 °C from early June to early September. Heavy rainfall was noted from early June to early July, and daily precipitation exceeding 100 mm was observed four times.

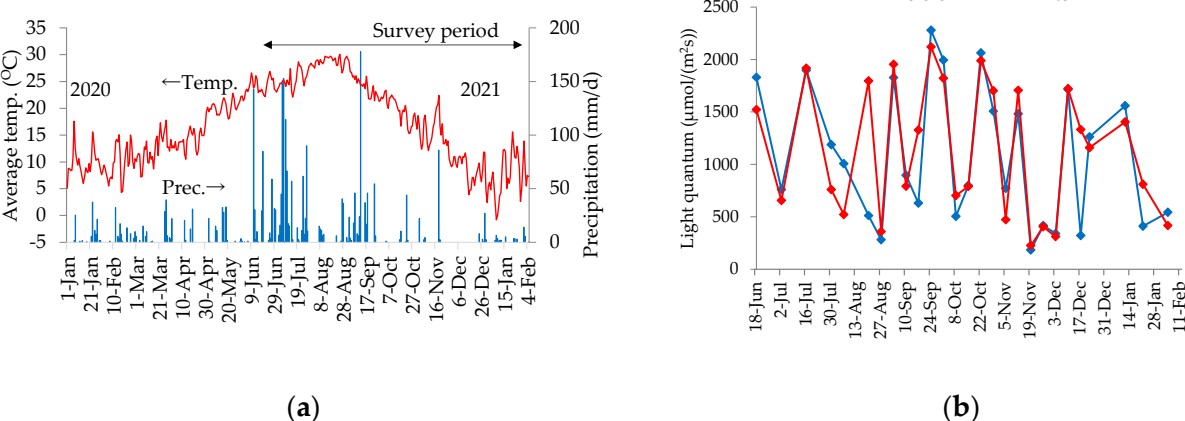

(**a**)                                                                                    (**b**)

**Figure 5.** (**a**) Average temperature and precipitation; (**b**) changes in light quanta.

Figure 5b shows changes in open-air light quanta at the site. Light quanta before and after sampling are shown separately. All measurements were taken in the morning between 9:00 a.m. and 12:00 p.m. Light quanta on clear days and in September were high.

### 4.3. Changes in Water Quality Parameters

Changes in water temperature are shown in Figure 6. The average water temperatures in all sections were above 25 °C from 3 July to 18 September and below 25 °C from 25 September to 5 February (the last day of measurement). There were no clear outliers in the water temperatures of the sections, and the overall changes in water temperatures were similar, with less than 0.3 °C difference in mean values.

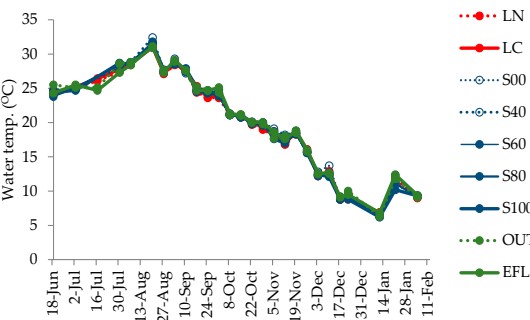

**Figure 6.** Changes in water temperature.

The changes in pH are shown by category in Figure 7a–c. OUT had high pH values in summer. Even though the upper limit of 7.5 was exceeded only on two occasions (31 July and 11 September) in OUT, it was possible to reproduce the situation of pH exceeding the upper limit in summer without shading. The highest pH was below 7.5 in the open air for both LN and S00. The seasonal trend of pH was not clear except for OUT. However, from the beginning of July to the end of September, there was a large difference in pH between the unshaded sections (S00 and OUT) and the corresponding shaded sections (S100 and EFL) in the small and entire shaded sections, indicating the effect of shading on pH reduction. This difference was not observed in the large sections.

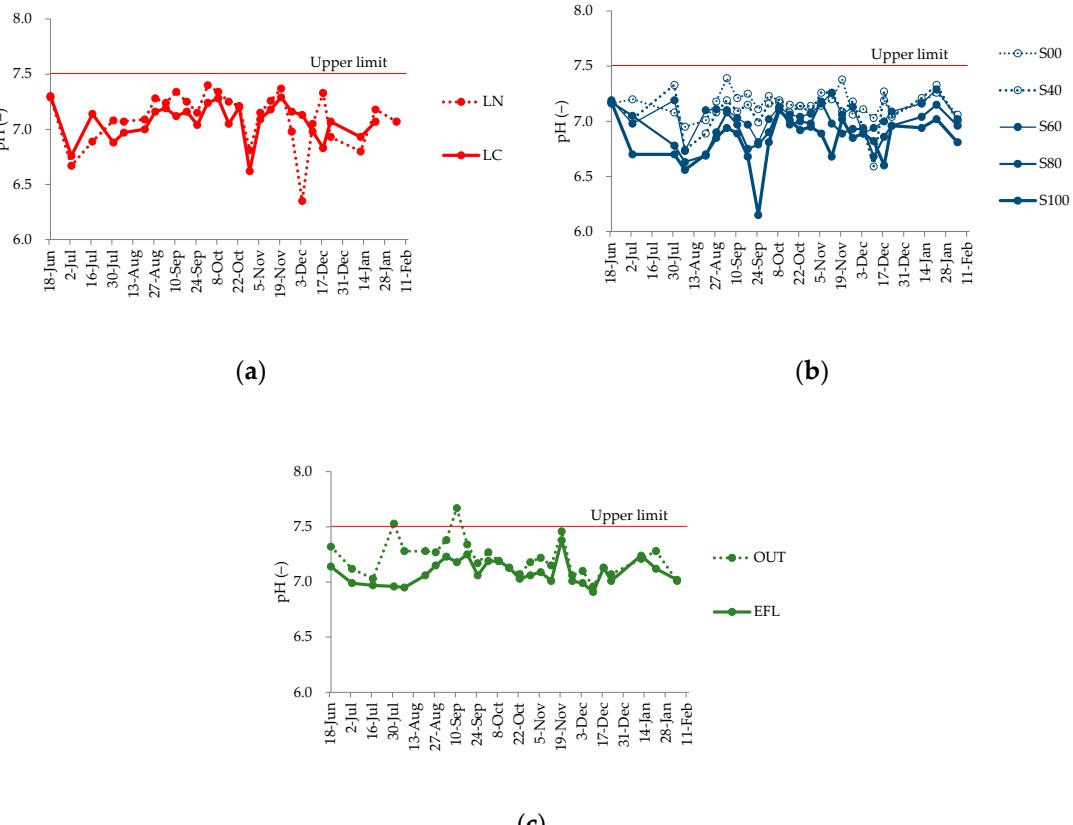

**Figure 7.** Changes in pH: (**a**) large section; (**b**) small section; (**c**) entire section.

The ECs were approximately 150 μS/cm until the beginning of July, exceeded 1100 μS/cm in the end of August and then decreased to and stabilized at approximately 500 μS/cm after the middle of September. No difference was noted between sections. The ORPs showed no clear trends, their values ranging from 100 to 400 mV (indicated values). The figures are omitted.

The trends of DOs are shown in Figure 8a. DO was low in summer and high in winter. This is due to the increase in the amount of DO caused by the decrease in water temperature. Table 2 shows that DOs in the shaded areas are generally lower than those in the unshaded areas. In other words, photosynthesis was inhibited by shading. In particular, DO in S100 was low from early to late November.

The trends of chlorophyll are shown in Figure 8b. Chlorophyll concentration was approximately 10 μg/L until the end of October, but spiked to levels exceeding 10 μg/L (up to 100 μg/L) thereafter. Whereas pH was high in summer, chlorophyll was high after autumn.

The trends of IC are shown in Figure 8c. Due to equipment malfunction, only data up to November 27 are shown. Despite two outliers, the values were generally in the range of 2 to 8 mg-C/L, and there was no significant difference among the sections.

Total nitrogen was generally in the range of 2 to 6 mg-N/L, and there was no clear trend. Total phosphorus was approximately 0.1 to 0.2 mg-P/L, and there was no clear trend as well (figures not shown). According to the national effluent standards for the protection of the living environment (lakes and marshes) in Japan, the permissible limit for total nitrogen and total phosphorus is 1 mg-N/L and 0.1 mg-P/L, respectively, which means that there is sufficient total nitrogen and total phosphorus to cause eutrophication.

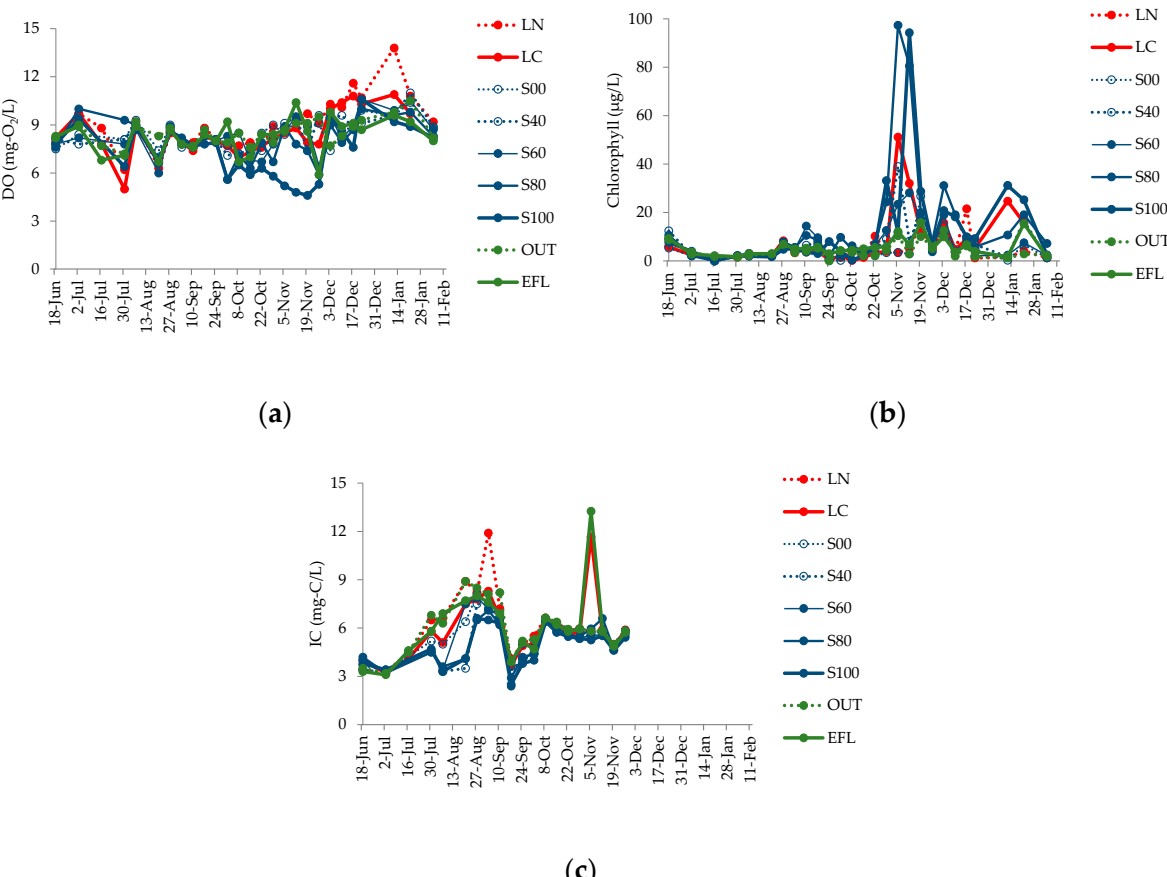

**Figure 8.** Changes in (**a**) DO; (**b**) chlorophyll; (**c**) IC.

## 5. Discussion

### 5.1. Relationship between Shading Effect and Alkalization Suppression Effect

We formulated the following hypothesis: if the shading effect were high, algal growth (chlorophyll production) would be reduced, the decrease in IC due to photosynthesis would be inhibited, and alkalization would be suppressed.

The average chlorophyll and IC levels for each section are shown in Figure 9a,b. As a general trend, the average chlorophyll levels were higher (IC levels were lower) in the shaded sections than the unshaded sections. This result runs counter to our hypothesis. In our previous water tank experiments, the average chlorophyll level was lower (IC level was higher) in the shaded area than the unshaded area [14]. Considering the nature of blue-green algae, which surface under dark conditions and settle when exposed to strong light [25,26], it is possible that algae in the unshaded section proliferated under direct sunlight, but settled thereafter. Therefore, when the surface water of each section was sampled, chlorophyll in the unshaded section may have been missed, resulting in the relatively low value. In the water tank described in our previous paper [14], even if algae settled in the unshaded section, it would be easy to obtain representative water quality parameters because of the limited depth of the tank.

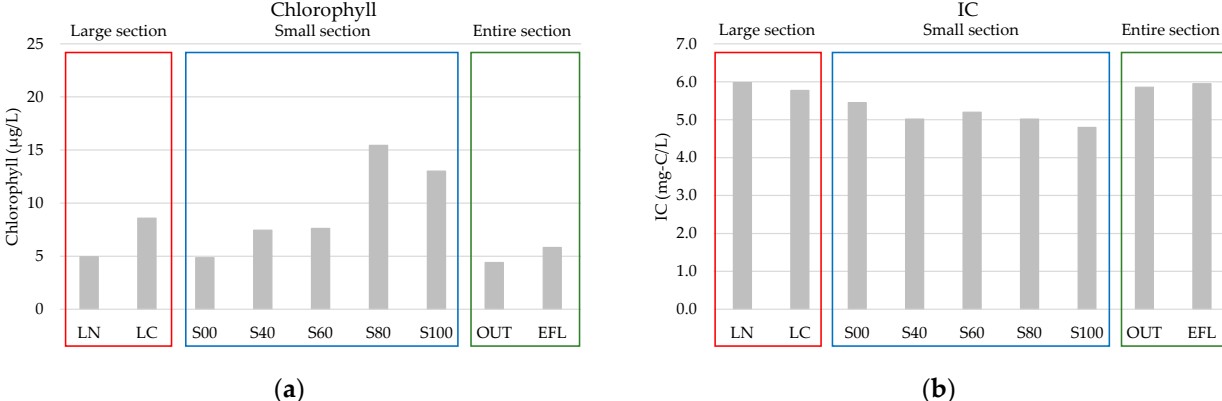

**Figure 9.** (**a**) Average chlorophyll level for each section; (**b**) average IC level for each section.

The average pH for each section is shown in Figure 10. The sections were clustered in a relatively small area, and the water samples were collected on the same day, differing only in the presence or absence of shading. Therefore, the water quality data of each section sampled on the same day could be paired. As pH is the logarithm of the reciprocal of hydrogen ion activity, it is an ordinal scale. Therefore, the Wilcoxon signed-rank test, a non-parametric statistical hypothesis test used to compare two related samples, was performed.

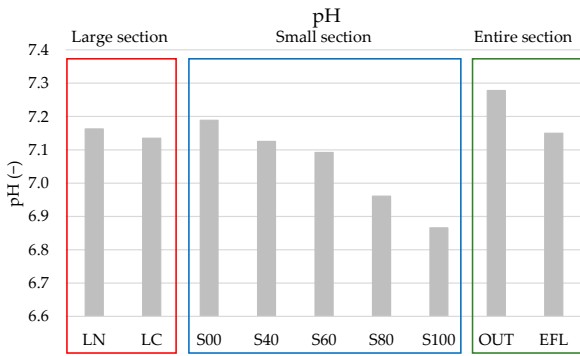

**Figure 10.** Average pH for each section.

In the large section, the mean pH value of LC was lower than that of LN, but the difference was not significant (degrees of freedom = 52, $p = 0.12$). In the entire section, the mean pH value of EFL (pH = 7.15) was lower than that of OUT (pH = 7.28), and the difference was significant (degrees of freedom = 52, $p < 0.0001$, effect size $d$ = combination with low EFL/all combinations = 22/27 = 0.81). The Friedman test was performed for the five groups of small sections, and a significant difference was found between some groups ($p < 0.0001$). Next, the matrix of $p$-values was obtained by the Wilcoxon signed-rank test, comparing five groups ($n = 27$, 10 combinations). The $p^*$-values after Bonferroni correction were calculated by multiplying the $p$-values by the number of combinations (Table 3). There was no significant difference between S00 and S40 ($p^* = 0.238$), but there was a significant difference between S00 and S60 ($p^* = 0.012$), S80 and S100 ($p^* < 0.001$). A clear decrease in pH was observed with more than 60% shading.

**Table 3.** *p\**-values after Bonferroni correction and *effect sizes d* for pH comparison of small sections.

|  | S00 | S40 | S60 | S80 | S100 |
|---|---|---|---|---|---|
| S00 | – | – | *0.70* | *0.96* | *0.96* |
| S40 | 0.238 | – | – | *0.93* | *0.96* |
| S60 | 0.012 | 0.256 | – | *0.89* | *0.96* |
| S80 | <0.001 | <0.001 | <0.001 | – | *0.74* |
| S100 | <0.001 | <0.001 | <0.001 | 0.029 | – |

These results indicate that there is no relationship between the shading effect and the decrease in algal growth (increase in IC), but there is a clear relationship between the shading effect and the suppression of alkalization.

*5.2. Problems, Prospects, Relevance of This Study*

The trends of chlorophyll and IC were found to run counter to the hypothesis. Since the lower part of the sections is open, even if algae proliferate, the algae cannot be collected when they settle, so chlorophyll and IC cannot be assessed accurately. It is necessary to measure algal growth at every depth, not only at the surface layer.

No significant difference in pH was observed in the large sections. Surface water sampling points were located at the edges of each section. Therefore, the collected samples would not be representative when the experimental scale was increased. Water in the sections should be mixed before collection.

The following are the prospects of this study. As the shading material was installed in 2020, we will continue observations and accumulate effluent pH data for an extended period. By doing so, we will be able to confirm whether the shading material significantly decreases pH. If the lifetime of the shading material is known, the cost savings for chemical treatment can be evaluated. If the pH of the effluent can be evaluated by changing the shading area of the entire RRR, the required shading area for the target pH can be estimated.

The results of this study can be used as primary data to estimate the use of shading materials as a countermeasure against the alkalization of water in ponds and regulating reservoirs.

**6. Conclusions**

We conducted a pilot-scale shading experiment in a rainwater regulating reservoir, and evaluated the effect of shading on pH. pH decreased from 7.28 to 7.15 when 3% of the total area of the rainwater regulating reservoir was shaded. In addition, a clear decrease in pH was observed with more than 60% shading.

**Author Contributions:** Conceptualization, H.A. and U.M.; methodology, K.N.; formal analysis, H.A.; investigation, H.A.; writing—original draft preparation, H.A.; writing—review and editing, H.A.; supervision, H.A. and U.M.; funding acquisition, U.M. All authors have read and agreed to the published version of the manuscript.

**Funding:** This research was funded by Nagasaki City as a consignment study (20200603). The APC was funded by the Institute of Integrated Sciences and Technology, Nagasaki University.

**Data Availability Statement:** Data are available upon reasonable request to the corresponding author.

**Acknowledgments:** The authors would like to thank Zhang Jiayu, Shintaro Morinaga and Yuki Suematsu (Faculty of Environmental Science, Nagasaki University) for help in sampling and measurements.

**Conflicts of Interest:** The authors declare no conflict of interest. The funders had no role in the design of the study; the collection, analyses or interpretation of data; the writing of the manuscript or the decision to publish the results.

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
