# Peer review of "Suppression of Alkalization in Rainwater Regulating Reservoir by Shading on a Pilot Scale"

_water, doi:10.3390/w13182557_

Round 1

Reviewer 1 Report

Please mention the type of algae that are formed and the water reservoir area.

line 62 - what type of combustible we are talking about?

line 77 - what type of sampler you used to collect the water?

line 102 - explain how you measured the ph at several levels.

Figure 1, please specify where the Upper limit is.

I can not understand what type of layers you are talking about. In the beginning, I was with the idea that were layered in deep, but now I think it is places of the reservoir. Please clarify that.

line 148 to 154 - please refer to if there is a water flux in the zone and if it is great or no. Because of high water flux, it will mix several waters with different pH.

Line 170 please put a figure of the water sampling or a photo.

I can not see values from SP: Final Effluent, I am mistaken?

Reviewer 2 Report

Abstract- this should be improved in accordance with the main findings.

Introduction:

Row 73 - please detail what is „treated leachate”.
The paragraph (rows 81-95) should be moved to the introduction.
Figure 2 - the authors are kindly requested to insert the error bars in the graphs.
Materials and methods:
It would have been interesting if the study was performed over a longer period to see if similar results are obtained.

The statistical methods that were used, should be also presented.

Row 178 - „After the measurements, 100 mL polyethylene bottles were filled with water and stored.” - Please mention the time and temperature for storage.

Results:
For the readers' convenience it t is commendable to insert the figures closer to the paragraphs where the discussion of the results is presented. 
The authors should consider discussing the values obtained for the chlorophyll (Table 2).

Conclusions

This section should be in accordance with the main findings presented in section 5. As it is presented in the manuscript, the conclusions are not consistent with the results.

Reviewer 3 Report

The study aims to follow the pH reduction effect of shading on a pilot-scale shading experiment in rainwater regulating reservoir. The (study) results are very promising, well organized. However, the comparison of the results was done in a univariate way (variable by variable). I recommend a multivariate comparison introducing chemometric tools for exploration (Principal component analysis) or clustering (Kmeans or others) using the results from Table 2. A multivariate comparison is highly advised to highlight or evaluate the variation of those parameters in a correct way. Then the comparison between large, small, and entire sections will be based on the entire parameters. 

Author Response

Thank you for your constructive and kind comments. We believe that the method you suggest will allow for meaningful comparisons between sections. I would like to study your proposed method and make every effort to preserve the environment. However, since I need to learn from scratch, I would like to do a simple statistical analysis for this paper.

Round 2

Reviewer 2 Report

The authors adequately responded to the reviewer's comments.